# Privacy-Preserving Logistic Regression Training with A Faster Gradient Variant

## Abstract

Logistic regression training over encrypted data has been an attractive idea to security concerns for years. In this paper, we propose a faster gradient variant called `quadratic gradient` to implement logistic regression training in a homomorphic encryption domain, the core of which can be seen as an extension of the simplified fixed Hessian [5]. We enhance Nesterov's accelerated gradient (NAG) and Adaptive Gradient Algorithm (Adagrad) respectively with this gradient variant and evaluate the enhanced algorithms on several datasets. Experimental results show that the enhanced methods have a state-of-the-art performance in convergence speed compared to the naive first-order gradient methods. We then adopt the enhanced NAG method to implement homomorphic logistic regression training and obtain a comparable result by only 3 iterations.

## 1 Introduction

Logistic regression (LR) is a widely used classification technology especially in medical risk assessment due to its simplicity but powerful performance. Given a person's healthcare data related to a certain disease, we can train an LR model capable of telling whether or not this person is likely to develop this disease. However, such personal health information is highly private to individuals. The privacy concern, therefore, becomes a major obstacle for individuals to share their biomedical data. The most secure solution is to encrypt the data into ciphertexts first by Homomorphic Encryption (HE) and then securely outsource the ciphertexts to the cloud, without allowing the cloud to access the data directly. iDASH is an annual competition that aims to call for implementing interesting cryptographic schemes in a biological context. Since 2014, iDASH has included the theme of genomics and biomedical privacy. The third track of the 2017 iDASH competition and the second track of the 2018 iDASH competition were both to develop homomorphic-encryption-based solutions for building an LR model over encrypted data.

Several studies on logistic regression models are based on homomorphic encryption. Aono et al. [2] only used an additive HE scheme and left some of the challenging HE computations to a trusted client. Kim et al. [14] discussed the problem of performing LR training in an encrypted environment. They used the full batch gradient descent in the training process and the least square method to get the approximation of the sigmoid function. In the iDASH 2017 competition, Bonte and Vercauteren [5], Kim et al. [12], Chen et al. [6], and Crawford et al. [8] all investigated the same problem that Kim et al. [14] studied. In the iDASH competition of 2018, Kim et al. [13] and Blatt et al. [3] further worked on it for an efficient packing and semi-parallel algorithm. The papers most relevant to this work are [5] and [12]. Bonte and Vercauteren [5] developed a practical algorithm called the simplified fixed Hessian (SFH) method. Our study complements their work and adopts the ciphertext packing technique proposed by Kim et al. [12] for efficient homomorphic computation.

Submitted to 36th Conference on Neural Information Processing Systems (NeurIPS 2022). Do not distribute.

Our specific contributions in this paper are as follows:

1. We propose a new gradient variant, `quadratic gradient`, which can unite the first-order gradient method and the second-order Newton's method as one.

2. We develop two enhanced gradient methods by equipping the original methods with `quadratic gradient`. The resulting methods show a better performance in the convergence speed.

3. We adopt the enhanced Nesterov's accelerated gradient to implement privacy-preserving logistical regression training, to our best knowledge, which seems to be the best candidate without compromising much on computation and storage.

## 2 Preliminaries

We adopt the square brackets "[ ]" to denote the index of a vector or matrix element in what follows. For example, for a vector $v \in \mathbb{R}^{(n)}$ and a matrix $M \in \mathbb{R}^{m \times n}$, $v[i]$ or $v_{[i]}$ means the $i$-th element of vector $v$ and $M[i][j]$ or $M_{[i][j]}$ the $j$-th element in the $i$-th row of $M$.

### 2.1 Fully Homomorphic Encryption

Fully Homomorphic Encryption (FHE) is a type of cryptographic scheme that can be used to compute an arbitrary number of additions and multiplications directly on the encrypted data. It was not until 2009 that Gentry constructed the first FHE scheme via a bootstrapping operation [9]. FHE schemes themselves are computationally time-consuming; the choice of dataset encoding matters likewise to the efficiency. In addition to these two limits, how to manage the magnitude of plaintext [11] also contributes to the slowdown. Cheon et al. [7] proposed a method to construct an HE scheme with a `rescaling` procedure which could eliminate this technical bottleneck effectively. We adopt their open-source implementation `HEAAN` while implementing our homomorphic LR algorithms. It is inevitable to pack a vector of multiple plaintexts into a single ciphertext for yielding a better amortized time of homomorphic computation. `HEAAN` supports a parallel technique (aka `SIMD`) to pack multiple numbers in a single polynomial by virtue of the Chinese Remainder Theorem and provides rotation operation on plaintext slots. The underlying HE scheme in `HEAAN` is well described in [12, 14, 10].

### 2.2 Database Encoding Method

Kim et al. [12] devised an efficient and promising database-encoding method by using `SIMD` technique, which could make full use of the computation and storage resources. Suppose that a database has a training dataset consisting of $n$ samples with $(1 + d)$ covariates, they packed the training dataset $Z$ into a single ciphertext in a row-by-row manner:

Using this encoding scheme, we can manipulate the data matrix $Z$ by performing HE operations on the ciphertext $Enc[Z]$, with the help of only three HE operations - rotation, addition and multiplication. For example, if we want the first column of $Enc[Z]$ alone and filter out the other columns, we can

design a constant matrix $F$ consisting of ones in the first column and zeros in the rest columns and then multiply $Enc[Z]$ by $Enc[F]$, obtaining the resulting ciphertext $Enc[Z_p]$:

$$Enc[F] \otimes Enc[Z] = Enc[Z_p] \quad \text{(where "$\otimes$" means the component-wise HE multiplication)}$$

$$= Enc \begin{bmatrix} 1 & 0 & \dots & 0 \\ 1 & 0 & \dots & 0 \\ \vdots & \vdots & \ddots & \vdots \\ 1 & 0 & \dots & 0 \end{bmatrix} \otimes Enc \begin{bmatrix} z_{[1][0]} & z_{[1][1]} & \cdots & z_{[1][d]} \\ z_{[2][0]} & z_{[2][1]} & \cdots & z_{[2][d]} \\ \vdots & \vdots & \ddots & \vdots \\ z_{[n][0]} & z_{[n][1]} & \cdots & z_{[n][d]} \end{bmatrix} = Enc \begin{bmatrix} z_{[1][0]} & 0 & \dots & 0 \\ z_{[2][0]} & 0 & \dots & 0 \\ \vdots & \vdots & \ddots & \vdots \\ z_{[n][0]} & 0 & \dots & 0 \end{bmatrix}.$$

Han et al. [10] introduced several operations to manipulate the ciphertexts, such as a procedure named "SumColVec" to compute the summation of the columns of a matrix. By dint of these basic operations, more complex calculations such as computing the gradients in logistic regression models are feasible.

## 2.3 Logistic Regression

Logistic regression is widely used in binary classification tasks to infer whether a binary-valued variable belongs to a certain class or not. LR can be generalized from linear regression [15] by mapping the whole real line ($\boldsymbol{\beta}^T\mathbf{x}$) to $(0, 1)$ via the sigmoid function $\sigma(z) = 1/(1+\exp(-z))$, where the vector $\boldsymbol{\beta} \in \mathbb{R}^{(1+d)}$ is the main parameter of LR and the vector $\mathbf{x} = (1, x_1, \dots, x_d) \in \mathbb{R}^{(1+d)}$ the input covariate. Thus logistic regression can be formulated with the class label $y \in \{\pm 1\}$ as follows:

$$\Pr(y = +1|\mathbf{x}, \boldsymbol{\beta}) = \sigma(\boldsymbol{\beta}^T\mathbf{x}) \qquad = \frac{1}{1 + e^{-\boldsymbol{\beta}^T\mathbf{x}}},$$

$$\Pr(y = -1|\mathbf{x}, \boldsymbol{\beta}) = 1 - \sigma(\boldsymbol{\beta}^T\mathbf{x}) \quad = \frac{1}{1 + e^{+\boldsymbol{\beta}^T\mathbf{x}}}.$$

LR sets a threshold (usually $0.5$) and compares its output with it to decide the resulting class label.

The logistic regression problem can be transformed into an optimization problem that seeks a parameter $\boldsymbol{\beta}$ to maximize $L(\boldsymbol{\beta}) = \prod_{i=1}^{n} \Pr(y_i|\mathbf{x}_i, \boldsymbol{\beta})$ or its log-likelihood function $l(\boldsymbol{\beta})$ for convenience in the calculation:

$$l(\boldsymbol{\beta}) = \ln L(\boldsymbol{\beta}) = -\sum_{i=1}^{n} \ln(1 + e^{-y_i\boldsymbol{\beta}^T\mathbf{x}_i}),$$

where $n$ is the number of examples in the training dataset. LR does not have a closed form of maximizing $l(\boldsymbol{\beta})$ and two main methods are adopted to estimate the parameters of an LR model: (a) gradient descent method via the gradient; and (b) Newton's method by the Hessian matrix. The gradient and Hessian of the log-likelihood function $l(\boldsymbol{\beta})$ are given by, respectively:

$$\nabla_{\boldsymbol{\beta}} l(\boldsymbol{\beta}) = \sum_i (1 - \sigma(y_i\boldsymbol{\beta}^T\mathbf{x}_i))y_i\mathbf{x}_i,$$

$$\nabla_{\boldsymbol{\beta}}^2 l(\boldsymbol{\beta}) = \sum_i (y_i\mathbf{x}_i)(\sigma(y_i\boldsymbol{\beta}^T\mathbf{x}_i) - 1)\sigma(y_i\boldsymbol{\beta}^T\mathbf{x}_i)(y_i\mathbf{x}_i)$$

$$= X^T S X$$

where $S$ is a diagonal matrix with entries $S_{ii} = (\sigma(y_i\boldsymbol{\beta}^T\mathbf{x}_i) - 1)\sigma(y_i\boldsymbol{\beta}^T\mathbf{x}_i)$ and $X$ the dataset.

The log-likelihood function $l(\boldsymbol{\beta})$ of LR has at most a unique global maximum [1], where its gradient is zero. Newton's method is a second-order technique to numerically find the roots of a real-valued differentiable function, and thus can be used to solve the $\boldsymbol{\beta}$ in $\nabla_{\boldsymbol{\beta}} l(\boldsymbol{\beta}) = 0$ for LR.

## 3 Technical Details

It is quite time-consuming to compute the Hessian matrix and its inverse in Newton's method for each iteration. One way to limit this downside is to replace the varying Hessian with a fixed matrix $\bar{H}$. This novel technique is called the fixed Hessian Newton's method. Böhning and Lindsay [4] have shown that the convergence of Newton's method is guaranteed as long as $\bar{H} \leq \nabla_{\boldsymbol{\beta}}^2 l(\boldsymbol{\beta})$, where $\bar{H}$ is

a symmetric negative-definite matrix independent of $\boldsymbol{\beta}$ and "$\preceq$" denotes the Loewner ordering in the sense that the difference $\nabla_{\boldsymbol{\beta}}^2 l(\boldsymbol{\beta}) - \bar{H}$ is non-negative definite. With such a fixed Hessian matrix $\bar{H}$, the iteration for Newton's method can be simplified to:

$$\boldsymbol{\beta}_{t+1} = \boldsymbol{\beta}_t - \bar{H}^{-1}\nabla_{\boldsymbol{\beta}}l(\boldsymbol{\beta}).$$

Böhning and Lindsay also suggest the fixed matrix $\bar{H} = -\frac{1}{4}X^T X$ is a good lower bound for the Hessian of the log-likelihood function $l(\boldsymbol{\beta})$ in LR.

## 3.1 the Simplified Fixed Hessian method

Bonte and Vercauteren [5] simplify this lower bound $\bar{H}$ further due to the need for inverting the fixed Hessian in the encrypted domain. They replace the matrix $\bar{H}$ with a diagonal matrix $B$ whose diagonal elements are simply the sums of each row in $\bar{H}$. They also suggest a specific order of calculation to get $B$ more efficiently. Their new approximation $B$ of the fixed Hessian is:

$$B = \begin{bmatrix} \sum_{i=0}^{d} \bar{h}_{0i} & 0 & \cdots & 0 \\ 0 & \sum_{i=0}^{d} \bar{h}_{1i} & \cdots & 0 \\ \vdots & \vdots & \ddots & \vdots \\ 0 & 0 & \cdots & \sum_{i=0}^{d} \bar{h}_{di} \end{bmatrix},$$

where $\bar{h}_{ki}$ is the element of $\bar{H}$. This diagonal matrix $B$ is in a very simple form and can be obtained from $\bar{H}$ without much difficulty. The inverse of $B$ can be approximated in the encrypted form by means of computing the inverse of every diagonal element of $B$ via the iterative of Newton's method with an appropriate start value. Their simplified fixed Hessian method can be formulated as follows:

$$\begin{aligned}\boldsymbol{\beta}_{t+1} &= \boldsymbol{\beta}_t - B^{-1} \cdot \nabla_{\boldsymbol{\beta}}l(\boldsymbol{\beta}), \\ &= \boldsymbol{\beta}_t - \begin{bmatrix} b_{00} & 0 & \cdots & 0 \\ 0 & b_{11} & \cdots & 0 \\ \vdots & \vdots & \ddots & \vdots \\ 0 & 0 & \cdots & b_{dd} \end{bmatrix} \cdot \begin{bmatrix} \nabla_0 \\ \nabla_1 \\ \vdots \\ \nabla_d \end{bmatrix} = \boldsymbol{\beta}_t - \begin{bmatrix} b_{00} \cdot \nabla_0 \\ b_{11} \cdot \nabla_1 \\ \vdots \\ b_{dd} \cdot \nabla_d \end{bmatrix},\end{aligned}$$

where $b_{ii}$ is the reciprocal of $\sum_{i=0}^{d} \bar{h}_{0i}$ and $\nabla_i$ is the element of $\nabla_{\boldsymbol{\beta}}l(\boldsymbol{\beta})$.

Consider a special situation: if $b_{00}, \ldots, b_{dd}$ are all the same value $-\eta$ with $\eta > 0$, the iterative formula of the SFH method can be given as:

$$\boldsymbol{\beta}_{t+1} = \boldsymbol{\beta}_t - (-\eta) \cdot \begin{bmatrix} \nabla_0 \\ \nabla_1 \\ \vdots \\ \nabla_d \end{bmatrix} = \boldsymbol{\beta}_t + \eta \cdot \nabla_{\boldsymbol{\beta}}l(\boldsymbol{\beta}),$$

which is the same as the formula of the naive gradient $ascent$ method. Such coincident is just what the idea behind this work comes from: there is some relation between the Hessian matrix and the learning rate of the gradient (descent) method. We consider $b_{ii} \cdot \nabla_i$ as a new enhanced gradient variant's element and assign a new learning rate to it. As long as we ensure that this new learning rate decreases from a positive floating-point number greater than 1 (such as 2) to 1 in a bounded number of iteration steps, the fixed Hessian Newton's method guarantees the algorithm will converge eventually.

The SFH method proposed by Bonte and Vercauteren [5] has two limitations: (a) in the construction of the simplified fixed Hessian matrix, all entries in the symmetric matrix $\bar{H}$ need to be non-positive. For machine learning applications the datasets will be in advance normalized into the range [0,1], meeting the convergence condition of the SFH method. However, for other cases such as numerical optimization, it doesn't always hold; and (b) the simplified fixed Hessian matrix $B$ that Bonte and Vercauteren [5] constructed can still be singular, especially when the dataset is a high-dimensional sparse matrix, such as the datasets from the MNIST database. We complement their work by removing these limitations so as to generalize this simplified fixed Hessian to be invertible in any case and propose a faster gradient variant, which we term `quadratic gradient`.

## 3.2 Quadratic Gradient

Suppose that a differentiable scalar-valued function $F(\mathbf{x})$ has its gradient $\boldsymbol{g}$ and Hessian matrix $H$, with any matrix $\bar{H} \leq H$ in the Loewner ordering as follows:

$$\boldsymbol{g} = \begin{bmatrix} g_0 \\ g_1 \\ \vdots \\ g_d \end{bmatrix}, \quad H = \begin{bmatrix} \nabla^2_{00} & \nabla^2_{01} & \cdots & \nabla^2_{0d} \\ \nabla^2_{10} & \nabla^2_{11} & \cdots & \nabla^2_{1d} \\ \vdots & \vdots & \ddots & \vdots \\ \nabla^2_{d0} & \nabla^2_{d1} & \cdots & \nabla^2_{dd} \end{bmatrix}, \quad \bar{H} = \begin{bmatrix} \bar{h}_{00} & \bar{h}_{01} & \cdots & \bar{h}_{0d} \\ \bar{h}_{10} & \bar{h}_{11} & \cdots & \bar{h}_{1d} \\ \vdots & \vdots & \ddots & \vdots \\ \bar{h}_{d0} & \bar{h}_{d1} & \cdots & \bar{h}_{dd} \end{bmatrix},$$

where $\nabla^2_{ij} = \nabla^2_{ji} = \frac{\partial^2 F}{\partial x_i \partial x_j}$. We construct a new Hessian matrix $\tilde{B}$ as follows:

$$\tilde{B} = \begin{bmatrix} -\varepsilon - \sum_{i=0}^{d} |\bar{h}_{0i}| & 0 & \cdots & 0 \\ 0 & -\varepsilon - \sum_{i=0}^{d} |\bar{h}_{1i}| & \cdots & 0 \\ \vdots & \vdots & \ddots & \vdots \\ 0 & 0 & \cdots & -\varepsilon - \sum_{i=0}^{d} |\bar{h}_{di}| \end{bmatrix},$$

where $\varepsilon$ is a small positive constant to avoid division by zero (usually set to $1e-8$).

As long as $\tilde{B}$ satisfies the convergence condition of the above fixed Hessian method, $\tilde{B} \leq H$, we can use this approximation $\tilde{B}$ of the Hessian matrix as a lower bound. Since we already assume that $\bar{H} \leq H$, it will suffice to show that $\tilde{B} \leq \bar{H}$. We prove $\tilde{B} \leq \bar{H}$ in a similar way that [5] did.

**Lemma 1.** *Let $A \in \mathbb{R}^{n \times n}$ be a symmetric matrix, and let $B$ be the diagonal matrix whose diagonal entries $B_{kk} = -\varepsilon - \sum_i |A_{ki}|$ for $k = 1, \ldots, n$, then $B \leq A$.*

*Proof.* By definition of the Loewner ordering, we have to prove the difference matrix $C = A - B$ is non-negative definite, which means that all the eigenvalues of $C$ need to be non-negative. By construction of $C$ we have that $C_{ij} = A_{ij} + \varepsilon + \sum_{k=1}^{n} |A_{ik}|$ for $i = j$ and $C_{ij} = A_{ij}$ for $i \neq j$. By means of Gerschgorin's circle theorem, we can bound every eigenvalue $\lambda$ of $C$ in the sense that $|\lambda - C_{ii}| \leq \sum_{i \neq j} |C_{ij}|$ for some index $i \in \{1, 2, \ldots, n\}$. We conclude that $\lambda \geq A_{ii} + \varepsilon + |A_{ii}| \geq \varepsilon > 0$ for all eigenvalues $\lambda$ and thus that $B \leq A$. $\square$

**Definition 3.1** (`Quadratic Gradient`). *Given such a $\tilde{B}$ above, we define the quadratic gradient as $G = \bar{B} \cdot \boldsymbol{g}$ with a new learning rate $\eta$, where $\bar{B}$ is a diagonal matrix with diagonal entries $\bar{B}_{kk} = 1/|\tilde{B}_{kk}|$, and $\eta$ should be always no less than 1 and decrease to 1 in a limited number of iteration steps. Note that $G$ is still a column vector of the same size as the gradient $\boldsymbol{g}$. To maximize or minimize the function $F(\mathbf{x})$, we can use the iterative formulas: $\mathbf{x}_{k+1} = \mathbf{x}_k + \eta \cdot G$ or $\mathbf{x}_{k+1} = \mathbf{x}_k - \eta \cdot G$, just like the naive gradient.*

We point out here that $\bar{H}$ could be the Hessian matrix $H$ itself and $\tilde{B}$ further optimized to: $\tilde{B}_{kk} = \bar{h}_{kk} + |\bar{h}_{kk}| + \varepsilon - \sum_{i=0}^{d} |\bar{h}_{ki}|$. In our experiments, we use $\bar{H} = -\frac{1}{4} X^T X$ to construct our $\tilde{B}$.

## 3.3 Two Enhanced Methods

`Quadratic Gradient` can be used to enhance NAG and Adagrad.

NAG is a different variant of the momentum method to give the momentum term much more prescience. The iterative formulas of the gradient *ascent* method for NAG are as follows:

$$V_{t+1} = \boldsymbol{\beta}_t + \alpha_t \cdot \nabla J(\boldsymbol{\beta}_t), \tag{3}$$

$$\boldsymbol{\beta}_{t+1} = (1 - \gamma_t) \cdot V_{t+1} + \gamma_t \cdot V_t, \tag{4}$$

where $V_{t+1}$ is the intermediate variable used for updating the final weight $\boldsymbol{\beta}_{t+1}$ and $\gamma_t \in (0, 1)$ is a smoothing parameter of moving average to evaluate the gradient at an approximate future position [12]. The enhanced NAG is to replace (3) with $V_{t+1} = \boldsymbol{\beta}_t + (1 + \alpha_t) \cdot G$. Our enhanced NAG method is described in Algorithm 1 .

Adagrad is a gradient-based algorithm suitable for dealing with sparse data. The updated operations of Adagrad and its quadratic-gradient version, for every parameter $\boldsymbol{\beta}_{[i]}$ at each iteration step $t$, are as

---

**Algorithm 1** The enhanced Nesterov's accelerated gradient method

---

**Input:** training dataset $X \in \mathbb{R}^{n \times (1+d)}$; training label $Y \in \mathbb{R}^{n \times 1}$; and the number $\kappa$ of iterations;
**Output:** the parameter vector $V \in \mathbb{R}^{(1+d)}$

1: Set $\bar{H} \leftarrow -\frac{1}{4} X^T X$                  $\triangleright \bar{H} \in \mathbb{R}^{(1+d) \times (1+d)}$
2: Set $V \leftarrow \mathbf{0}, W \leftarrow \mathbf{0}, \bar{B} \leftarrow \mathbf{0}$       $\triangleright V \in \mathbb{R}^{(1+d)}, W \in \mathbb{R}^{(1+d)}, \bar{B} \in \mathbb{R}^{(1+d) \times (1+d)}$
3: **for** $i := 0$ to $d$ **do**
4:      $\bar{B}[i][i] \leftarrow \varepsilon$               $\triangleright \varepsilon$ is a small positive constant such as $1e-8$
5:      **for** $j := 0$ to $d$ **do**
6:          $\bar{B}[i][i] \leftarrow \bar{B}[i][i] + |\bar{H}[i][j]|$
7:      **end for**
8: **end for**
9: Set $\alpha_0 \leftarrow 0.01, \alpha_1 \leftarrow 0.5 \times (1 + \sqrt{1 + 4 \times \alpha_0^2})$
10: **for** $count := 1$ to $\kappa$ **do**
11:      Set $Z \leftarrow \mathbf{0}$           $\triangleright Z \in \mathbb{R}^n$ is the inputs for sigmoid function
12:      **for** $i := 1$ to $n$ **do**
13:          **for** $j := 0$ to $d$ **do**
14:              $Z[i] \leftarrow Z[i] + Y[i] \times V[j] \times X[i][j]$
15:          **end for**
16:      **end for**
17:      Set $\boldsymbol{\sigma} \leftarrow \mathbf{0}$         $\triangleright \boldsymbol{\sigma} \in \mathbb{R}^n$ is to store the outputs of the sigmoid function
18:      **for** $i := 1$ to $n$ **do**
19:          $\boldsymbol{\sigma}[i] \leftarrow 1/(1 + \exp(-Z[i]))$
20:      **end for**
21:      Set $\boldsymbol{g} \leftarrow \mathbf{0}$
22:      **for** $j := 0$ to $d$ **do**
23:          **for** $i := 1$ to $n$ **do**
24:              $\boldsymbol{g}[j] \leftarrow \boldsymbol{g}[j] + (1 - \boldsymbol{\sigma}[i]) \times Y[i] \times X[i][j]$
25:          **end for**
26:      **end for**
27:      Set $G \leftarrow \mathbf{0}$
28:      **for** $j := 0$ to $d$ **do**
29:          $G[j] \leftarrow \bar{B}[j][j] \times \boldsymbol{g}[j]$
30:      **end for**
31:      Set $\eta \leftarrow (1 - \alpha_0)/\alpha_1, \gamma \leftarrow 1/(n \times count)$       $\triangleright n$ is the size of training data
32:      **for** $j := 0$ to $d$ **do**
33:          $w_{temp} \leftarrow V[j] + (1 + \gamma) \times G[j]$
34:          $V[j] \leftarrow (1 - \eta) \times w_{temp} + \eta \times W[j]$
35:          $W[j] \leftarrow w_{temp}$
36:      **end for**
37:      $\alpha_0 \leftarrow \alpha_1, \alpha_1 \leftarrow 0.5 \times (1 + \sqrt{1 + 4 \times \alpha_0^2})$
38: **end for**
39: **return** $V$

---

follows, respectively:

$$\boldsymbol{\beta}_{[i]}^{(t+1)} = \boldsymbol{\beta}_{[i]}^{(t)} - \frac{\eta}{\varepsilon + \sqrt{\sum_{k=1}^{t} \boldsymbol{g}_{[i]}^{(t)} \cdot \boldsymbol{g}_{[i]}^{(t)}}} \cdot \boldsymbol{g}_{[i]}^{(t)},$$

$$\boldsymbol{\beta}_{[i]}^{(t+1)} = \boldsymbol{\beta}_{[i]}^{(t)} - \frac{1 + \eta}{\varepsilon + \sqrt{\sum_{k=1}^{t} G_{[i]}^{(t)} \cdot G_{[i]}^{(t)}}} \cdot G_{[i]}^{(t)}.$$

**Performance Evaluation** We evaluate the performance of various algorithms in the clear using the Python programming language on the same desktop computer with an Intel Core CPU G640 at 1.60 GHz and 7.3 GB RAM. Since our focus is on how fast the algorithms converge in the training phase, the loss function, maximum likelihood estimation (MLE), is selected as the only indicator. We evaluate four algorithms, NAG, Adagrad, and their quadratic-gradient versions (denoted as Enhanced NAG and Enhanced Adagrad, respectively) on the datasets that Kim et al. [12] adopted: the iDASH genomic dataset (iDASH), the Myocardial Infarction dataset from Edinburgh (Edinburgh), Low Birth weight Study (lbw), Nhanes III (nhanes3), Prostate Cancer study (pcs), and Umaru Impact Study datasets (uis). The genomic dataset is provided by the third task in the iDASH competition of 2017, which consists of 1579 records. Each record has 103 binary genotypes and a binary phenotype indicating if the patient has cancer. The other five datasets all have a single binary dependent variable. Figures 1 and 2 show that except for the enhanced Adagrad method on the iDASH genomic dataset our enhanced methods all converge faster than their original ones in other cases.

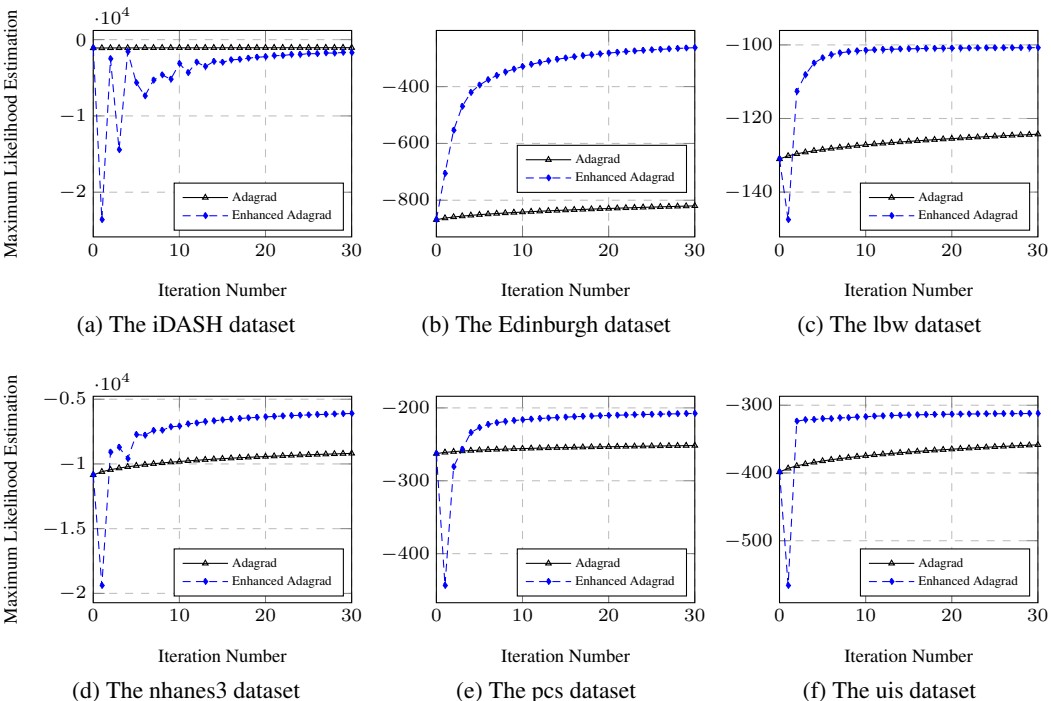

Figure 1: Training results in the clear for Adagrad and Enhanced Adagrad

In all the Python experiments, the time to calculate the $\bar{B}$ in quadratic gradient $G$ before running the iterations and the time to run each iteration for various algorithms are negligible (few seconds).

## 4 Privacy-preserving Logistic Regression Training

Adagrad method is not a practical solution for homomorphic LR due to its frequent inversion operations. It seems plausible that the enhanced NAG is probably the best choice for privacy-preserving LR training. We adopt the enhanced NAG method to implement privacy-preserving logistic regression training. The difficulty in applying the quadratic gradient is to invert the diagonal

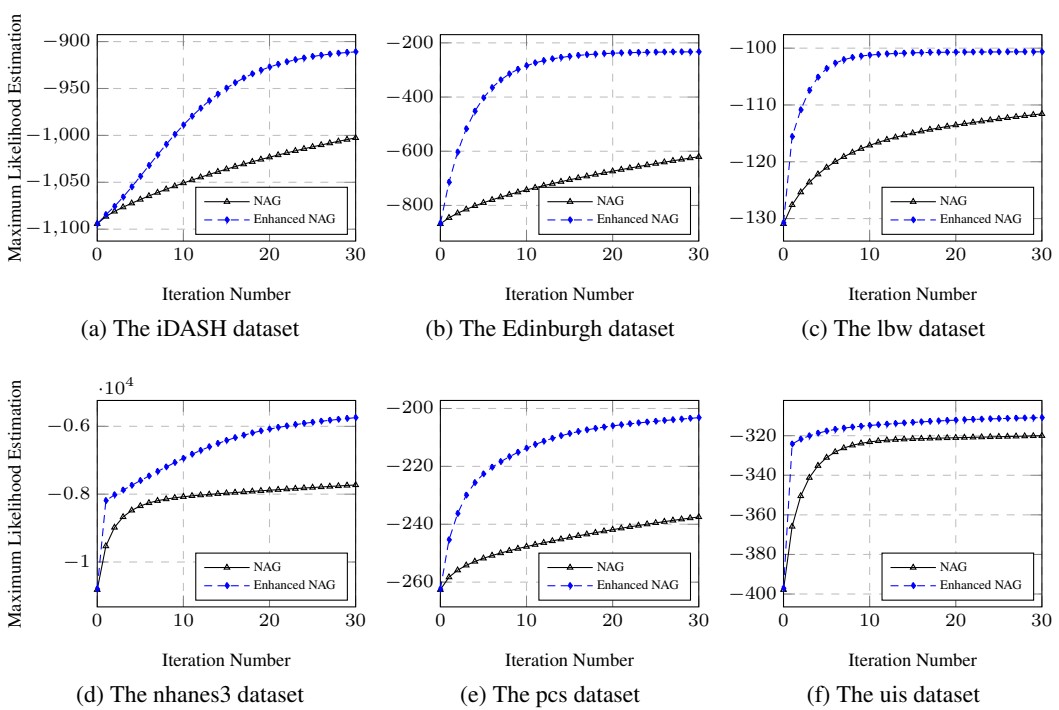

Figure 2: Training results in the clear for NAG and Enhanced NAG

matrix $\tilde{B}$ in order to obtain $\bar{B}$. We leave the computation of matrix $\bar{B}$ to data owner and let the data owner upload the ciphertext encrypting the $\bar{B}$ to the cloud. Since data owner has to prepare the dataset and normalize it, it would also be practicable for the data owner to calculate the $\bar{B}$ owing to no leaking of sensitive data information.

Privacy-preserving logistic regression training via homomorphic encryption technique faces a difficult dilemma that no homomorphic schemes are capable of directly calculating the sigmoid function in the LR model. A common solution is to replace the sigmoid function with a polynomial approximation by using the widely adopted least square method. We can call a function named "`polyfit(·)`" in the Python package Numpy to fit the polynomial in a least-square sense. We adopt the degree 5 polynomial approximation $g(x)$ by which Kim et al. [12] used the least square approach to approximate the sigmoid function over the domain $[-8, 8]$: $g(x) = 0.5 + 0.19131 \cdot x - 0.0045963 \cdot x^3 + 0.0000412332 \cdot x^5$ .

Given the training dataset $X \in \mathbb{R}^{n \times (1+d)}$ and training label $Y \in \mathbb{R}^{n \times 1}$, we adopt the same method that Kim et al. [12] used to encrypt the data matrix consisting of the training data combined with training-label information into a single ciphertext $\text{ct}_Z$. The weight vector $\beta^{(0)}$ consisting of zeros and the diagonal elements of $\bar{B}$ are copied $n$ times to form two matrices. The data owner then encrypt the two matrices into two ciphertexts $\text{ct}_\beta^{(0)}$ and $\text{ct}_{\bar{B}}$, respectively. The ciphertexts $\text{ct}_Z$, $\text{ct}_\beta^{(0)}$ and $\text{ct}_{\bar{B}}$ are as follows:

$$
X = \begin{bmatrix} 1 & x_{11} & \dots & x_{1d} \\ 1 & x_{21} & \dots & x_{2d} \\ \vdots & \vdots & \ddots & \vdots \\ 1 & x_{n1} & \dots & x_{nd} \end{bmatrix}, Y = \begin{bmatrix} y_1 \\ y_2 \\ \vdots \\ y_n \end{bmatrix}, \text{ct}_Z = Enc \begin{bmatrix} y_1 & y_1 x_{11} & \dots & y_1 x_{1d} \\ y_2 & y_2 x_{21} & \dots & y_2 x_{2d} \\ \vdots & \vdots & \ddots & \vdots \\ y_n & y_n x_{n1} & \dots & y_n x_{nd} \end{bmatrix},
$$

$$
\text{ct}_\beta^{(0)} = Enc \begin{bmatrix} \beta_0^{(0)} & \beta_1^{(0)} & \dots & \beta_d^{(0)} \\ \beta_0^{(0)} & \beta_1^{(0)} & \dots & \beta_d^{(0)} \\ \vdots & \vdots & \ddots & \vdots \\ \beta_0^{(0)} & \beta_1^{(0)} & \dots & \beta_d^{(0)} \end{bmatrix}, \text{ct}_{\bar{B}} = Enc \begin{bmatrix} \bar{B}_{[0][0]} & \bar{B}_{[1][1]} & \dots & \bar{B}_{[d][d]} \\ \bar{B}_{[0][0]} & \bar{B}_{[1][1]} & \dots & \bar{B}_{[d][d]} \\ \vdots & \vdots & \ddots & \vdots \\ \bar{B}_{[0][0]} & \bar{B}_{[1][1]} & \dots & \bar{B}_{[d][d]} \end{bmatrix},
$$

where $\bar{B}_{[i][i]}$ is the diagonal element of $\bar{B}$ that is built from $-\frac{1}{4}X^T X$ .

The pulbic cloud takes the three ciphertexts $\text{ct}_Z$, $\text{ct}_\beta^{(0)}$ and $\text{ct}_{\bar{B}}$ and evaluates the enhanced NAG algorithm to find a decent weight vector by updating the vector $\text{ct}_\beta^{(0)}$. Refer to [12] for a detailed description about how to calculate the gradient by HE programming.

**Implementation**    We implement the enhanced NAG based on HE with the library `HEAAN`. The C++ source code is publicly available at `https://anonymous.4open.science/r/IDASH2017-245B`. All the experiments on the ciphertexts were conducted on a public cloud with 32 vCPUs and 64 GB RAM.

For a fair comparison with [12], we utilized the same 10-fold cross-validation (CV) technique on the same iDASH dataset consisting of 1579 samples with 18 features and the same 5-fold CV technique on the other five datasets. Like [12], We consider the average accuracy and the Area Under the Curve (AUC) as the main indicators. Tables 1 and 2 show the two experiment results, respectively. The two tables also provide the average evaluation running time for each iteration. We adopt the same packing method that Kim et al. [12] proposed and hence our solution has similar storage of ciphertexts to [12] with some extra ciphertexts to encrypt the $\bar{B}$.

The parameters of `HEAAN` we set are same to [12]: $logN = 16$, $logQ = 1200$, $logp = 30$, $slots = 32768$, which ensure the security level $\lambda = 80$. Refer [12] for the details of these parameters. Since our enhanced NAG method need to consume more modulus to preserve the precision of $\bar{B}$, we use $logp = 60$ to encrypt the matrix $\bar{B}$ and thus only can perform 3 iterations of the enhanced NAG method. Yet despite only 3 iterations, our enhanced NAG method still produces a comparable result.

Table 1: Implementation Results for iDASH datasets with 10-fold CV

| Dataset | Sample Num | Feature Num | Method | deg $g$ | Iter Num | Learn Time (min) | Accuracy (%) | AUC |
|---------|-----------|-------------|--------|---------|----------|------------------|--------------|-----|
| iDASH | 1579 | 18 | Ours | 5 | 3 | 5.53 | 53.69 | 0.678 |
|        |      |    | [12] | 5 | 7 | 6.07 | 62.87 | 0.689 |

Table 2: Implementation Results for other datasets with 5-fold CV

| Dataset | Sample Num | Feature Num | Method | deg $g$ | Iter Num | Learn Time (min) | Accuracy (%) | AUC |
|---------|-----------|-------------|--------|---------|----------|------------------|--------------|-----|
| Edinburgh | 1253 | 9 | Ours | 5 | 3 | 0.6 | 84.4 | 0.853 |
|           |      |   | [12] | 5 | 7 | 3.6 | 91.04 | 0.958 |
| lbw | 189 | 9 | Ours | 5 | 3 | 0.5 | 69.19 | 0.619 |
|     |     |   | [12] | 5 | 7 | 3.3 | 69.19 | 0.689 |
| nhanes3 | 15649 | 15 | Ours | 5 | 3 | 5.5 | 79.23 | 0.490 |
|         |       |    | [12] | 5 | 7 | 7.3 | 79.22 | 0.717 |
| pcs | 379 | 9 | Ours | 5 | 3 | 0.6 | 65.33 | 0.721 |
|     |     |   | [12] | 5 | 7 | 3.5 | 68.27 | 0.740 |
| uis | 575 | 8 | Ours | 5 | 3 | 0.6 | 74.43 | 0.598 |
|     |     |   | [12] | 5 | 7 | 3.5 | 74.44 | 0.603 |

# 5    Conclusion

In this paper, we proposed a faster gradient variant called `quadratic gradient`, and implemented the quadratic-gradient version of NAG in the encrypted domain to train the logistic regression model.

The quadratic gradient presented in this work can be constructed from the Hessian matrix directly, and thus somehow integrates the second-order Newton's method and the first-order gradient (descent) method together. There is a good chance that quadratic gradient could accelerate other gradient methods such as RMSprop and Adam, which is an open future work.

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

# A    Appendix

Optionally include extra information (complete proofs, additional experiments and plots) in the appendix. This section will often be part of the supplemental material.

