# OpenReview forum: "Privacy-Preserving Logistic Regression Training with A Faster Gradient Variant"
_NeurIPS.cc/2022/Conference — NeurIPS 2022 Submitted_

### Official Review · Reviewer_o6pq · 2022-07-05

**Rating:** 3
**Confidence:** 5
**Soundness:** 2 fair
**Presentation:** 2 fair
**Contribution:** 2 fair

**Summary:**

This study suggests an enhanced gradient method on logistic regression with homomorphic encryption (HE)
The contents of the paper generally follows (Kim et al., 2018), so for who read the paper the contents are familiar.
The others modified existing algorithm by replacing learning rate with quadratic gradient, which is a variant of simplified fixed Hessian.
Experimental results show that the proposed method converges better than existing method in (Kim et al., 2018).

**Questions:**

1. Parameter setting is an important issue when using HE, because parameter search is extremely unfriendly with HE. In algorithm 1, the authors set $\alpha_{t+1}=(1+\sqrt{1+4\alpha_{t}^2})/2$, which is different to the setting in (Kim et al., 2018), which is $\alpha_t=10/(t+1)$. Can the parameter choice be justified?

2. In the experiments, iDash dataset showed different results than other datasets. Can it be explained?

3. Can the results with $\lambda=128$ be given?

4. In table 2, the auc score of Ours on nhanes3 is too low. Is it a typo?

**Limitations:**

The authors have not address the limitations of their work. Because the study is about privacy-preserving logistic regression, it seems to have a positive societal impact.

As mentioned above, the biggest limitation of their work is that there is no significant improvement compared to the existing methods.

**Strengths And Weaknesses:**

Strengths
- In table 2, It is empirically shown that the proposed method can achieve similar accuracy compared to (Kim et al., 2018) with much less computation time. (the ratio of the computation time is bigger than 7/3 for most datasets, which seems because bigger $logq=60$ was used for the proposed method)
- By transferring the role of computing $\tilde{B}$ to the data owner, the proposed method does not require more multiplicative depth compared to (Kim et al., 2018).
- The paper seems clear and easy to follow.

Weaknesses
- The novelty seems rather shallow; (Kim et al., 2020) has already proposed an approximate Hessian which is very similar to quadratic gradient method. The authors should check the paper.
- Algorithm 1 seems rather meaningless. The algorithm should incorporate operations on ciphertexts, which deal with ciphertext packing.

(Kim et al., 2020)  Kim, M., Lee, J., Ohno-Machado, L., & Jiang, X. (2019). Secure and differentially private logistic regression for horizontally distributed data. IEEE Transactions on Information Forensics and Security, 15, 695-710.

---

> ### Author Response · Authors · 2022-08-01
> **Response to reviewer  o6pq**
>
> We would like to thank the reviewers for their input and appreciate their comments.
>
> The approximate Hessian proposed by (Kim et al., 2020) is very similar to the quadratic gradient but different in some ways: (a) they only used the diagonal elements of the fixed Hessian to build the approximate Hessian, together with a regularization term. They have to decide the parameter in this term. The quadratic gradient takes advantage of all the elements of the fixed Hessian. (b) (Kim et al., 2020) probably cannot be applied to numerical optimization. The underlying theory of quadratic gradient has been proved to meet the converge condition of the fixed Hessian method, so it might be probably used for general optimization problems (future work needs to be done to justify this claim).
>
> Algorithm 1 is the pseudo-code to implement the enhanced NAG method in the clear. The algorithm in the clear is really the same as  the one on ciphertexts, from the logistic perspective. Moreover, the baseline work presented a detailed description of the algorithm incorporating operations on ciphertexts. We have no new contributions to the ciphertext packing and the ciphertext-computation process, so our first mission is to clearly represent the algorithm logistic behind the enhanced NAG.
>
>
> We implement our privacy-preserving logistic regression training based on the open source code of the baseline method. Our learning-rate setting $\alpha_{t+1} = (1 + \sqrt{1 + 4\alpha_t^2})/2  $ is based on the line 128 in their source code at: https://github.com/kimandrik/IDASH2017/blob/master/IDASH2017/src/TestGD.cpp
>   (the first author changed the link from https://github.com/kimandrik/HEML to https://github.com/kimandrik/IDASH2017).
> The NAG method has many variants and the one that the baseline method used doesn't need to tune the learning rate very much. Although the authors claimed in their paper that they chose a learning rate $\alpha_t = 10/(t + 1)$, we found no such setting in their open source code.
>
>
> We don't know the reason why the enhanced Adagrad method had a different result in the iDash dataset. Our guess is that the enhanced Adagrad method might not perform well at the first several iterations because it has no enough prior knowledge to get a  good learning-rate setting (the raw Adagrad has a less serious problem with this) but catches up with the raw Adagrad after it has enough quadratic gradients to obtain its learning rate. It might be probably the case that the raw Adagrad performs very well in the iDash dataset and when the enhanced Adagrad method finally catches up, there is no much room for both methods to improve in their performance.
> We should not have included the enhanced Adagrad method, which is not related very much to the topic of this paper and whose performance we don't fully understand. Maybe we should leave it to future work.
>
> For a fair comparison, we adopt the same parameters of the HE setting as the baseline work.  The security parameter $\lambda$ is determined by the parameters $\log N$ and $\log Q$ according to the security estimator of (Albreht et al. 2015). (Kim et al. 2018) derived a lower-bound on the ring dimension as $N \ge \frac{\lambda + 110}{7.2} \cdot \log Q$ to get $\lambda$-bit security level. Since HEAAN uses the complex number to facilitate the FFT computation, it needs another $N$ slots to store the complex conjugates. The formula is actually $N \ge 2 \cdot \frac{\lambda + 110}{7.2} \cdot \log Q$ for HEAAN.
> The results with $\lambda = 128 $ can be obtained if we set $\log N = 17$ and $\log Q = 1200$.
>
>
> In table 2, that the auc score of Ours on nhanes3 is too low is probably because the nhanes3 is a too large dataset and only 3 iterations is not enough to achieve a decent auc score. We doubted it was a typo before and therefore implemented the bootstrapping in our code to perform more iterations, just to see the further outcomes. After many experiments, the results show that enough iterations enable the enhanced method to have a comparable auc score compared to the baseline method. Here is one running result: https://anonymous.4open.science/r/IDASH2017-245B/IDASH2017/Debug/No.vCPU2_RAM16GB_Datasetnhanes3_kdeg5_fold5_numIter17_nohup.out
>                          So we don't think it is a typo.
>
>
>
> (Albreht et al. 2015) M. R. Albrecht, R. Player, and S. Scott. On the concrete hardness of learning with errors. Journal of Mathematical Cryptology, 9(3):169{203, 2015
>
> (Kim et al. 2018) Kim, M., Song, Y., Wang, S., Xia, Y., and Jiang, X. (2018b). Secure logistic regression based
> 262 on homomorphic encryption: Design and evaluation. JMIR medical informatics, 6(2):e19.

---

### Official Review · Reviewer_2fQY · 2022-07-11

**Rating:** 5
**Confidence:** 4
**Soundness:** 2 fair
**Presentation:** 3 good
**Contribution:** 2 fair

**Summary:**

This paper presents a faster gradient variant to implement logistic regression training in a homomorphic encryption domain. The results show the new optimization method could speed up the convergence of training on small datasets.

**Questions:**

1. In Figure 1(a), why the convergence speed of Enhanced Adagrad is worse than the raw Adagrad?
2. I am curious about the consumption of memory in the HE domain, it will be better if the authors include a comparison between the proposed method and the baseline method.
3. I like the idea of the second-order optimization and wonder whether the SFH method could be applied to more general NNs like CNN, LSTM or Transformer?


**Limitations:**

Yes, the authors have included the limitations in line 119, which includes (a) the numerical condition of the SFH method and (b) the singular problem for some high-dimensional sparse matrix datasets.

**Strengths And Weaknesses:**

Strength:
1. This paper is well-written and clearly organized, including sufficient theoretic analysis of the proposed Simplified Fixed Hessian method. I like such a solid line of work.
2. The targeted topic – privacy-preserving domain of logistic regression is very promising and deserves insightful hard work to resolve the application of homomorphic encryption.
3. The results of convergence speed are clear and convincing.

Weakness:
1. The experiments are somewhat not enough, considering the lack of experimental support for the claim in line 44 “without compromising much on computation and storage.” It will be better if a detailed comparison of computation and storage is presented.
2. The algorithm may lack of generalization to some large-scale datasets, considering the calculation of $\bar{\math{B}}$ needs to walk through the whole dataset, which is quite resource-hungry
3. Lack of further analysis of the results of the experiment, such as in Figure 1(a), why the convergence speed of Enhanced Adagrad is worse than the raw Adagrad.

---

> ### Author Response · Authors · 2022-08-01
> **Response to reviewer 2fQY**
>
> We would like to thank the reviewers for their input and appreciate their comments.
>
> Our method and the baseline method have very similar computation processes and our method adopts the ciphertext packing technique proposed by the baseline work. Therefore, the two works have very similar consumption of memory in the HE domain.
>
> For some large-scale datasets such as the MNIST dataset, the calculation of $\tilde B$ indeed needs to walk through the whole dataset.  However, in these cases, we usually adopt the mini-batch version of the NAG method rather than the full batch version and partition the large-scale dataset into multiple same-size batches. The mini-batch version of the enhanced NAG method would be an open future work.
>
> We don't know the reason why the convergence speed of Enhanced Adagrad is worse than the raw Adagrad. Our guess is that the enhanced Adagrad method might not perform well at the first several iterations because it has no enough prior knowledge to get a  good learning-rate setting (the raw Adagrad has a less serious problem with this) but catches up with the raw Adagrad after it has enough quadratic gradients to obtain a better learning rate. It might be probably the case that the raw Adagrad performs very well in the iDash dataset and when the enhanced Adagrad method finally catches up, there is no much room for both methods to improve in their performance.
>
>
> The SFH method, as an extension of the fixed Hessian method, has one main drawback that it cannot be used on some datasets like the MNIST datasets, in which case the simplified fixed Hessian is singular (The fixed Hessian method also has this weakness). This is one reason that the SFH method, as well as the fixed Hessian method, might not be applied to more general NNs.  Another important reason is that it might be difficult or even impossible to find a "fixed"  good lower bound of the Hessian matrix for the SFH method. On the other hand, the quadratic gradient can be constructed from the Hessian matrix directly, which doesn't rely on finding a fixed replaced matrix, and can be invertible in any case. This suggests that quadratic gradient might be able to be applied to more general NNs training like CNN, LSTM or Transformer.

---

> > ### Comment · Reviewer_2fQY · 2022-08-08
> > **Feedback**
> >
> > Thanks for the meaningful response. And after reading my colleague's comments, I decided to maintain my scores.

---

> > > ### Author Response · Authors · 2022-08-08
> > > **I am glad to help with the questions in my paper.**
> > >
> > > I am glad to help with the questions in my paper.
> > > And I am very grateful for the time you and other reviewers spent reading my work.

---

### Official Review · Reviewer_mJWs · 2022-07-13

**Rating:** 4
**Confidence:** 3
**Soundness:** 3 good
**Presentation:** 2 fair
**Contribution:** 2 fair

**Summary:**

This paper studied the problem of privacy-preserving logistic training on encrypted data. The quadratic gradient is proposed to implement logistic regression training in the homomorphic encryption domain. The proposed method can be seen as an extension of the simplified fixed Hessian and an enhancement of the Adaptive Gradient Algorithm (Adagrad). The implemented homomorphic logistic regression training obtains a comparable result by only 3 iterations.

**Questions:**

-

**Limitations:**

-

**Strengths And Weaknesses:**

Strengths:

This paper studies an important problem. Privacy-preserving logistic regression training is one popular method to protect the training data privacy, especially when logistic regression is ubiquitous in the real world.
The paper provides extensive experimental details and implemented codes that are helpful for reproducibility.

Weakness:

The writing is not clear to distinguish between the proposed methods and previous methods. Most of section 3 are previous works, instead of proposed methods. For example, section 3.1 about the Simplified Fixed Hessian method is actually from previous works. It is better to write it in a background section. Section 3.2 seems to be the proposed method. Thus, it should be highlighted and clarified using diagrams, and clear descriptions. More importantly, it is not clear to know the motivation why the authors propose a quadratic gradient. What are the observations and insights that we have to using quadratic gradient?

Typos: for example: lines 183-184. Ower-->owner

More comparisons are needed. Although the authors mentioned [1], it is required to have a result comparison with [1] since the techniques are applicable to logistic regression training. It is also better to compare with [2].

The results are not robust or generalized. For example, the enhanced method in the iDash dataset of figure 1 is not robust and may be worse than the baseline method. It would be better to explain more about the reason. Otherwise, it may show that the proposed method is not generalized to other datasets. Also, in table 2, the proposed method has worse accuracy than the baseline method on the Edinburgh dataset.

[1] Han, K., Hong, S., Cheon, J., & Park, D. (2019). Logistic regression on homomorphic encrypted data at scale. In Proceedings of the AAAI conference on artificial intelligence (pp. 9466–9471).

[2] Bergamaschi, F., Halevi, S., Halevi, T., & Hunt, H. (2019). Homomorphic training of 30,000 logistic regression models. In International Conference on Applied Cryptography and Network Security (pp. 592–611).

---

> ### Author Response · Authors · 2022-08-01
> **Response to reviewer mJWs**
>
> We would like to thank the reviewers for their input and appreciate their comments.
>
>
> The motivation why we proposed such a gradient variant is that quadratic gradient might unite the first-order gradient (descent) method and the second-order Newton's method, probably substituting and superseding the line-search method used in Newton's method. We think that the two enhanced methods might be super-quadratic algorithms (we might be wrong but we believe so) although we cannot prove it now.
> We proposed the quadratic gradient with the hope of constructing something that possesses both the merits of the gradient descent method and Newton's method.  We think that quadratic gradient can enhance Newton's method and fill the gap between the gradient methods and Newton's method.
>
>
> We did try to make some comparisons with the work [1]. However, [1] used the mini-batch version of NAG while we adopt the full batch version of NAG. Therefore, It might be difficult to design experiments to fairly compare the two different types of NAG methods. We would like to leave the mini-batch version of the enhanced NAG method to future work, in which we of course would like to take [1] as the baseline work. Python experiments have already shown that the enhanced NAG outperforms the original NAG even in the mini-batch version.  Our work and our baseline work are both based on a single model, whereas [2] is not. Therefore, it may be inappropriate to compare our work with [2].
>
>
> It is true that the enhanced method in the iDash dataset of figure 1 is not robust and may be worse than the baseline method. we don't know the reason. Our guess is that the enhanced Adagrad method might not perform well at the first several iterations because it has no enough prior knowledge to get a  good learning-rate setting (the raw Adagrad has a less serious problem with this) but catches up with the raw Adagrad after it has enough quadratic gradients to obtain a better learning rate. It might be probably the case that the raw Adagrad performs very well in the iDash dataset and when the enhanced Adagrad method finally catches up, there is no much room for both methods to improve in their performance.
>
>
> The enhanced NAG method has worse accuracy on the Edinburgh dataset probably because it can only be performed for 3 iterations while the baseline method was performed for 7 iterations.

---

### Meta-Review · Area_Chair_ugGu · 2022-08-26

**Recommendation:** Reject
**Confidence:** Certain

**Metareview:**

Reviewers remained concerned about the novelty of the contribution, about the extent/limitations of experiments/comparisons to other methods, as well as about the fact that the method does not seem to outperform competitors in certain cases.

**Award:**

No

---

### Decision · Program_Chairs · 2022-09-14

Reject